# What Keeps Social Entrepreneurs Happy? Exploring Personality, Work Design, External Support, and Social Impact as Resources of Social Entrepreneurs' Mental Well-Being

**Philipp Kruse** [1,*] **, Eleanor Meda Chipeta** [2] **and Imke Ueberschär** [1]

1 Faculty of Psychology, Technical University Dresden, 01069 Dresden, Germany
2 Wits Business School, University of the Witwatersrand, 2 St Davids Place, Parktown, Johannesburg 2193, South Africa
* Correspondence: philipp.kruse@tu-dresden.de

**Abstract:** Social entrepreneurship (SE) is a new form of entrepreneurship dedicated to the creation of social value for its beneficiaries, either as a for-profit or not-for-profit enterprise. While, over the years, research has yielded notable insights regarding, e.g., social entrepreneurial nascence and motivations or contextual factors (dis-)favoring SE activity, not much is known about the resources social entrepreneurs have to maintain their mental well-being (MWB), which is essential for successfully accomplishing their social missions. The current study takes a psychological view, identifies four resource clusters (personality, work design, external support, and provision of social impact), and integrates these to empirically explore their predictive values for job-specific and general MWB. Building on a representative sample of South African social entrepreneurs from Gauteng and Limpopo Provinces, we apply structural equation modeling and find positive effects on social entrepreneur's MWB in all resource clusters. Moreover, comparing for-profit and not-for profit social entrepreneurs yields differences in the levels and mechanisms of their MWB resources. Despite notable limitations, such as using cross-sectional data and a limited sample generalizability, our work offers the first framework shedding light on social entrepreneurs' MWB-resources that can serve as a basis for future research and help SE-support programs to sustainably promote social entrepreneurs' MWB.

**Keywords:** social entrepreneur; well-being; job satisfaction; resources; personality; work design; support; social impact; business model differences

## 1. Introduction

From an economic point of view, high levels of entrepreneurship are desirable, as entrepreneurs drive innovation, create jobs, and thus contribute to productivity and growth in increasingly dynamic and competitive global markets (see Van Praag et al. [1] for a review). However, in the face of mounting social problems, such as poverty, inequality, and social marginalization, researchers and practitioners also acknowledge the potential of entrepreneurs to make a positive contribution to society, i.e., create social value [2,3]. An entrepreneurial form particularly dedicated to combining entrepreneurial behavior and innovativeness with the aspiration to alleviate social problems is coined "social entrepreneurship" [4,5]. Social enterprises (SEs) are driven by a social mission such as traditional non-profit organizations (NPOs). However, what makes them unique is that they apply innovative entrepreneurial means to reach their goals. While the landscape of SEs is diverse [6] and persistently growing [7], two general forms can be identified. For-profit SEs build on an elaborated business model and blend the fulfilment of their social mission, i.e., social-value creation with profit-making activities and the creation of financial value. Thus, they are often referred to as "hybrid" SEs [8,9]. In contrast, not-for-profit SEs do not generate their own income. However, they are also operating in a competitive

environment, as they use innovative ideas to mobilize limited resources, such as donations, public funding, or volunteers. Furthermore, unlike traditional NPOs, not-for-profit SEs usually operate in local communities and are rather small [10,11].

In the last 30 years, research on SEs has been persistently growing and still attracts scholarly interest from several different disciplines [12–14]. As a result, insightful discoveries regarding, e.g., social entrepreneurial nascence and motivation [8,15], the influence of culture and economic circumstances on SE-activity [3,16], or the richness of SE-activities [17], have been made. Furthermore, the (mainly) positive effects of SE activity on its beneficiaries and societies (generally referred to as social impact) is well-documented (see Rawhouser et al. [18] for an overview). However, to date, research remains silent on the impact of SE activity on *social entrepreneurs themselves*, particularly, their mental well-being (MWB), i.e., how happy they are. This is surprising, as MWB is not only central to the effective functioning of humans and their ability to perform [19] but also considered an important criterion entrepreneurs use to evaluate the success of their career [20]. Consequently, to sustain the socially beneficial effects of SE-activity, insights into what keeps social entrepreneurs happy are essential. However, not only do recent reviews show that scholarly work on the MWB of social entrepreneurs is almost non-existent [21,22] or rather concerned with how social entrepreneurial activity can contribute to societal well-being [23,24], but there is even a tendency to emphasize the risks and detrimental effects of SE activity on their MWB. To illustrate this point, the literature on mission drift, i.e., the risk that for-profit SEs fail to manage conflicting financial and social goals and end up losing their hybridity, stresses the enormously high challenges faced by social entrepreneurs (see Grimes et al. [25] for an overview). As a result, the risk of SEs to fail and the risk of social entrepreneurs to suffer from mental illnesses such as burnout are assumed to be even higher compared to commercially minded entrepreneurship [26,27].

In the face of (i) the scarcity of research on social entrepreneur's MWB and (ii) the dominance of challenges, stressors, and potential ill-being of social entrepreneurs in the limited literature, we focus on their resources by asking what keeps social entrepreneurs happy, i.e., what helps them to sustain their MWB despite the apparent challenges. Building on insights from psychology and general entrepreneurship research on MWB, the current study theoretically identified four central resources of social entrepreneurs (favorable personality traits, good work design, external support, and the successful provision of social impact) and empirically tested their influence on social entrepreneurs' job-specific and general MWB, using structural equation modeling (SEM). Furthermore, comparing for-profit and not-for-profit social entrepreneurs, we explored differences in the absolute levels of MWB and its four central resources, as well as underlying mechanisms. By doing so, we built on a sample of 347 social entrepreneurs from the South African Provinces of Gauteng and Limpopo, applying non-parametric statistical testing and multigroup SEM.

In the next sections, we elaborate more on the concept of social entrepreneurship and its different forms, conceptualize MWB, identify four central resources for social entrepreneurs' MWB, and derive our hypotheses and research question.

## 2. Social Entrepreneurs Mental Well-Being

### 2.1. Social Entrepreneurship as a New Form of Entrepreneurship

Dating back to the first scientific conceptualizations [28], the history of entrepreneurship has been and still is dominated by the perspective that entrepreneurial activity serves the purpose of generating profit for entrepreneurs and their stakeholders [29]. Consequently, entrepreneurial success metrics are primarily composed of economic criteria, such as productivity or revenue [30]. However, in the early 1980s, a book by Young [31] entitled *If Not for Profit, for What?* outlined the potentially beneficial effect of applying entrepreneurial and innovative means to the creation of social value. Going one step further, Waddock and Post [32] saw the potential to catalyze social change through entrepreneurship. Over the years, this perspective on social value creation through entrepreneurship alongside a growing political and societal demand that entrepreneurial activity should contribute more

to driving positive social change spurred more and more entrepreneurs to incorporate a social mission in their enterprise. This resulted in a new entrepreneurial form coined "social entrepreneurship".

Social enterprises (SEs) have the primary target to alleviate a social problem and thereby create social value. To illustrate, the Italian SE San Patrignano helps former prisoners and drug addicts, i.e., socially marginalized groups, to reintegrate into the labor market by employing them as dog trainers or bakers. The goods produced are sold to sustainably self-finance San Patrignano's concept [33]. Another example is the German Social Entrepreneurship Network SEND. This SE is dedicated to promoting the concept of social entrepreneurship in Germany and mobilizes resources for networking and lobby work to make politicians aware of the potential that SEs have to contribute to a fair and sustainable society. SEND's income is generated through public funding and donations [34]. These two social enterprises do not only exemplify the rich diversity of SE-activities; they also illustrate the two most common forms of social entrepreneurship. On the one hand, for-profit SEs such as San Patrignano fuse the creation of social and financial value. In other words, they build on an elaborated upon and financially profitable business model to fulfill their social mission enabling them to remain independent from external funding. Since for-profit social enterprises have two goals—social-value creation, which is primary, and financial value creation—they are often referred to as "hybrid enterprises". On the other hand, social enterprises can also operate in the non-profit sector like SEND does. The financial resources needed are not acquired through the production of their own goods or the provision of services. Not-for-profit SEs apply for public funding, try to acquire donations, and recruit volunteers. As these resources are limited, not-for-profit SEs operate in a competitive environment and apply entrepreneurial means to convince funding agencies and donors. However, despite the obvious difference regarding the financing of their activities, for-profit and not-for-profit SEs have a common core, including the centrality of their social mission, their innovativeness in social-value creation, their primary focus on local community impact, and the necessity to acquire their resources through action on competitive markets. Thus, to conclude, both SE forms can be considered two sides of the same medal trying to achieve similar targets with different means.

Over the years, interest in social entrepreneurship has become more apparent, as it was triggered by the increasing number of social enterprises worldwide. Scholars from diverse fields of study, e.g., economics, sociology, or psychology, have contributed to the development of social entrepreneurship into a scientifically institutionalized field with a growing number of top-tier publications in leading entrepreneurship outlets, as well as its own scholarly journals [7,12,35]. For example, previous research shed light on social entrepreneurial nascence [8,36], the role of values in SEs [15,37], and contextual factors (dis-)favoring SE-creation [16,38]. Moreover, SEs' contributions to create positive social change by helping their beneficiaries while making a social impact in their communities [2,18] is well-documented in developed [39,40] and developing nations [41,42]. Moreover, social entrepreneurship became a topic of interest for education scholars, as well those who reflect on skills needed for aspiring social entrepreneurs at university [43], link complex thinking and the social entrepreneurial process [44], and showcase negative effects when thinking of social value creation as an "easy task" [45]. This way, notable insights on which personal and knowledge-related assets help to facilitate social change and value creation through social entrepreneurship were provided. However, little is known about social entrepreneurs themselves, particularly their mental well-being (MWB), leaving a white spot on the scholarly social entrepreneurship map.

### 2.2. Conceptualizing Mental Well-Being

While well-being in general refers to the overall quality of life and individuals' functioning [46], mental well-being (MWB) is particularly concerned with their mental health, i.e., the extent to which people judge themselves as "happy". This positive view on MWB is rooted in a definition put forward by the World Health Organization (WHO) in 1950,

describing mental health as "a condition [ ... ] which enables the individual to achieve a satisfactory synthesis of his own potentially conflicting, instinctive drives; to form and maintain harmonious relations with others; and to participate in constructive changes in his social and physical environment". This perception that MWB is more than just the absence of mental ill-being but a distinct concept is not only theoretically sound but, as later research revealed, deeply rooted in biobehavioral systems. To exemplify, while (mental) well-being is triggered by the release of serotonin and oxytocin [47], (mental) ill-being is primarily linked to stress-related biomarkers and allostatic load [48]. Furthermore, in contrast to (mental) ill-being usually leading to avoidance behavior, (mental) well-being results in approach behavior [49].

Over the years, the concept of MWB was further elaborated upon in many disciplines, such as medicine, psychology, or public health. As a result, a number of different MWB-components emerged (see Warr [46] and Stephan, Rauch, and Hatak [22] for overviews). The current study is interested in the MWB of social entrepreneurs, i.e., individuals working in the same profession. Thus, the connection between job-specific and general MWB is of particular importance. Job-specific MWB encompasses the cognitive evaluation of one's work and the degree to which one is satisfied or dissatisfied with it. Consequently, in work sciences such as work psychology, job-specific MWB is often termed "job satisfaction" [50]. In contrast, general MWB is not domain specific, but a rather overarching cognitive evaluation of how satisfied people are with their overall life. This encompasses job satisfaction and other domain-specific forms, such as satisfaction with one's family or private life, which are combined. Accordingly, "life satisfaction" is largely used synonymously in work sciences [51]. As research in several different vocational groups and cultures has shown, job satisfaction is a notable influence on general MWB [52–54].

Through a social entrepreneurship lens, the effect of job satisfaction on general MWB should exist for two central reasons. First, like most entrepreneurs, social entrepreneurs work very long hours [55]; thus, their work plays a major role in their lives, and it seems reasonable to believe that social entrepreneurs "at least in part, generate [MWB] through their work" (p.293). Second, social entrepreneurial activity is primarily driven by the accomplishment of a social mission, i.e., social entrepreneurs' motivation to make the world a better place [56,57], a deeply rooted altruistic motivation [58], and corresponding values [15]. As these pro-social motives and values are considered more universal rather than job-specific [59], we take the view that the creation of social value at work is not only an important indicator for social entrepreneurs' job satisfaction but also affects their general MWB. Consequently, we derive the following hypothesis:

**H1.** *High levels of job satisfaction have a positive effect on social entrepreneurs' general MWB.*

In the following section, we dive deeper into the job satisfaction and general MWB resources of social entrepreneurs. After outlining (i) our understanding of the term "resources", we review the psychological and entrepreneurship literature and identify (ii) favorable personality traits, (iii) good work design, (vi) external support, and (v) the successful provision of social impact as central resources. These are then combined in our research model.

### 2.3. Resources for Job Satisfaction and Mental Well-Being of Social Entrepreneurs

(i) Our understanding of the term "resources"

While, in general, resources are acknowledged as entities that help to maintain effective functioning under challenging or adverse circumstances, depending on the context, different perspectives on the term can be remarked. In organizational science and strategic management, the resource-based view, as one of the most influential theories in this field [60], defines resources as "all assets, capabilities, organizational processes, firm attributes, information, knowledge, etc. controlled by a firm that enable the firm to conceive of and implement strategies that improve its efficiency and effectiveness" (cf. Priem and

Butler [61], p.27). Hereby, the focus lies on *firm*-related entities and outcomes. From a work psychology perspective, resources are defined as "physical, psychological, social, or organizational aspects of the job that may do any of the following: (a) be functional in achieving work goals; (b) reduce job demands at the associated physiological and psychological costs; (c) stimulate personal growth and development" (p. 501) [62]. This encompasses a more *person*-centered focus and puts emphasis on what is needed for an individual to successfully accomplish his/her goals at work. In entrepreneurship research, both perspectives are feasible and frequently applied (see Kellermanns et al. [63] and Stephan [21] for overviews), and taking either a more firm-related or person-centered perspective depends on individual study goals. As we chose to focus on the MWB of social entrepreneurs and their job satisfaction, i.e., genuinely *individual* and psychological variables, we decided on adopting a work psychology perspective and the respective understanding of resources. Thus, resources for individual MWB can be internal, i.e., originate from the person him/herself (e.g., advantageous personality traits), or external, i.e., rooted in the work itself (e.g., favorable work design or success at work) or the working context (e.g., external support) [64].

(ii) Personality

The personality of an individual reflects "the enduring set of traits and styles that he or she exhibits" ([65]; p. 4). Personality traits are not only a central guide for important life decisions such as occupational choice [66]; they also affect job satisfaction (see Judge and Larsen [67] for a theoretical review and Judge et al. [68] for a meta-analysis). One central overarching personality aspect related to job satisfaction in all jobs is the so-called person–job fit [69]. This concept expresses the extent to which personal characteristics (e.g., personality traits) and job demands match each other. The underlying logic is that a high person–job fit enables people to meet all work requirements, perform well, and leads to positive attitudes toward one's work. A comprehensive meta-analysis underpinned this notion and revealed high effects of person–job fit on job satisfaction [70]. Consequently, a high person–job fit, i.e., favorable personality traits matching the demands in social entrepreneurship, should positively affect social entrepreneurs' job satisfaction. Furthermore, research employing the job-demands–resources model [62], one of the most influential models on well- and ill-being at work, revealed that advantageous personality traits can act as a resource and increase individual commitment at work [71].

We take the view that trait core confidence, openness, and self-transcendence are favorable personality traits for social entrepreneurs and can serve as internal psychological resources for job satisfaction and MWB.

Trait core confidence (TCC) can be defined as the dispositional belief of individuals to accomplish certain tasks [72]. In contrast to self-efficacy, however, TCC is more general and not bound to the cognitive evaluation of specific situations. TCC is a generalized belief in one's ability to handle psychological distress and master challenges by applying active problem-solving strategies and taking control of problems. This seems particularly important for social entrepreneurs, as they face high levels of stress from trying to combine a social mission and innovative entrepreneurial behavior to acquire resources for the accomplishment of their mission [73]. Furthermore, for-profit SEs are confronted with the challenge of handling the ambiguity of their roles as a creator of social value and monetary income to sustainably finance their actions [8]. Drawing from general entrepreneurship research, Sergent et al. [74] found a notable reduction of psychological distress amongst entrepreneurs with a high TCC. Considering that a reasonable level of psychological distress at work is linked with higher job satisfaction and well-being, we derive the following hypothesis:

**H2.1.** *High levels of trait core confidence have a positive effect on social entrepreneurs' job satisfaction and general MWB.*

Following the definition of Schwartz [59], people with high levels of openness highly value free thinking, innovative actions, and new experiences, and they are not shy of

taking risks. All of these elements express a high person–job fit with a career as a social entrepreneur. Free thinking and innovativeness are helpful, as the combination of social-value creation and entrepreneurial means requires new and trail-blazing ideas [75]. Furthermore, the preparedness to take risks is of great importance, as social entrepreneurs are confronted with several sources of failure, encompassing failure to mobilize enough financial resources, failure to achieve the social mission, or failure to balance financial and social value creation [25]. As empirical studies amongst nascent SEs showed the beneficial effects of openness in the SE-job context [8,15], and openness as a personal value dimension is defined as a central life goal, i.e., not restricted to job-related attitudes [59], we consider it a resource for job satisfaction and general MWB:

**H2.2.** *High levels of openness have a positive effect on social entrepreneurs' job satisfaction and general MWB.*

Another personal value dimension that contributes to a high person–job fit and thus can be expected to be a resource of social entrepreneurs' job satisfaction and MWB is self-transcendence. Defined as the central life goal to take care of and benevolently help others, self-transcendence expresses a deeply rooted altruism and willingness to support other people; in other words, it is the desire to create social value [59]. As this is the central driver of social entrepreneurs' actions [58], job satisfaction and general MWB should benefit from high self-transcendence values. Empirically, there is growing evidence that self-transcendence and related constructs positively impact social entrepreneurial nascence [8] and behavior [76]. Thus, we derive the following hypothesis:

**H2.3.** *High levels of self-transcendence have a positive effect on social entrepreneurs' job satisfaction and general MWB.*

(iii) Work design

Generally speaking, work design refers to the organization and kinds of tasks at the workplace [77]. Research on work design can look back at a long and rich tradition spanning more than 100 years and unearth important insights into how good and effective work can be crafted [78]. One of the most influential theoretical models in the field is the Job Characteristics Model by Hackman and Oldham [79]. This model postulates five central elements of well-designed and satisfying working tasks. Task variety, task identity, and task significance describe to which extent one's work (i) encompasses different sets of actions and skills (variety); (ii) is considered "complete", i.e., enables workers to contribute to all steps of a good's production process (identity); and (iii) is perceived to make sense, i.e., yields a contribution for others (significance). Task autonomy refers to the possibility of workers to make their own decisions regarding the scheduling, location, or other aspects of their work, and task feedback signifies the possibility to evaluate one's successful completion of a working task based on information imparted by the task. The model states that, the higher all five elements are, the more satisfied workers will be. Empirically, these assumptions have been tested and confirmed in a variety of contexts and can be considered very reliable (see Humphrey et al. [80] for an overview and meta-analysis). Taking an entrepreneurship perspective, one key motivation for nascent entrepreneurs to start their own venture despite the apparent risk of failure is that, "by launching new ventures, entrepreneurs hope to secure the highly enriched work environments they so strongly crave" (p. 370) [81]. Following this rationale, the aspiration to found one's own enterprise is strongly linked to the perception of "bad" work design when working as an employee and the hope for "better" work design when working as an entrepreneur [22]. This, however, goes hand in hand with the task of actively designing one's own work (and later also the work for one's employees) in a self-regulated manner. Thus, entrepreneurs are not only entrepreneurs but also designers of their own work [81]. Regarding the aspects of good work design postulated in the Job Characteristics Model, particularly the positive effect of autonomy on entrepreneurial job satisfaction has been investigated

and supported. Moreover, task variety, task identity, task significance, and task feedback were studied and emerged as resources for entrepreneurs' MWB (see Stephan [21] for an overview). Despite the low number of studies in general entrepreneurship research and, to the best of our knowledge, there being no such study in the SE context, it seems reasonable to expect positive relationships for social entrepreneurs, as well. Furthermore, we take the view that, while work design has direct effects on job satisfaction, its effect on general MWB is mediated through job satisfaction. Following the model of distance and proximity by Kanfer [82], work design characteristics have a higher proximity to work-related MWB, i.e., job satisfaction, than the relatively distal criterion of general MWB. Thus, it can be expected that the effects on general MWB are mediated. This assumption received empirical backing considering that (i) the Kanfer framework has already shown its suitability in SE research [8,15,83] and (ii) correlations of work design elements are notably lower for general MWB indicators than for job satisfaction [80]. Consequently, we derive the following hypothesis:

**H3.** *A good work design, i.e., high levels of task variety, task identity, task significance, task autonomy, and task feedback, has a positive direct effect on social entrepreneurs' job satisfaction and a positive indirect effect on general MWB via job satisfaction.*

(iv) External support

Most jobs are not limited to the individual completion of tasks but are embedded in a social context, as interactions with supervisors and/or co-workers are required [84]. Thus, in addition to task-specific work design, social support, i.e., the extent to which a job provides an opportunity for getting advice and assistance from other people at one's workplace, is considered an important influence on job satisfaction. In fact, extending the job characteristics model with this component significantly increased the explanatory power regarding job satisfaction over all occupations [80], and social support is one of the best studied external resources for entrepreneurs [21]. From an SE perspective, the favorable influence of social support has been shown in SE nascence [8], and networking and peer-support are seen as essential to sustain the success of a social enterprise (see Perrini, Vurro and Costanzo [33] for a case study and conceptual model). The reasons for this lie, amongst others, in the opportunity to share experiences and best practices amongst peers and enter beneficial networks or get to know with investors [85]. Furthermore, social support is widely acknowledged as a resource for general MWB [86] and also as being important for entrepreneurs' MWB [21]. Social support is not limited to job-specific advice but also includes the opportunity to make friends and thereby contributes to life satisfaction [87]. As a result, we derive the following hypothesis:

**H4.1.** *High levels of social support have a positive effect on social entrepreneurs' job satisfaction and general MWB.*

Furthermore, drawing from Institutional Theory [88], so-called institutional support turned out to exert an influence on (social) entrepreneurial activity. The provision of money, educational, or other resources, e.g., by governments, creates a supportive and favorable ecosystem for entrepreneurs which has been shown to positively affect the success of enterprises in general [22] and social enterprises in particular [3,16]. Research suggests that a favorable work context is linked to higher job satisfaction and reduces the risk of mental health problems [80]. The latter finding is also in line with one of the most influential and empirically solid models on mental well- and ill-being, the Job Demands-Resources Model by Bakker and Demerouti [89,90]. The model's central assumption is that job-related resources, such as a favorable work environment through institutional support, can buffer job-related demands, such as the creation of social value through entrepreneurial means on a competitive market, and decrease the risk of mental ill-being. Thus, we expect that high levels of institutional support positively affect the job satisfaction and general MWB of social entrepreneurs and derive the following hypothesis:

**H4.2.** *High levels of institutional support have a positive effect on social entrepreneurs' job satisfaction and general MWB.*

(v) Provision of social impact

While the landscape of social enterprises is becoming increasingly diverse and a wide array of different approaches are being applied [6,17], the central target, namely creating social value and thereby contributing to positive social change in society, unites all social entrepreneurs. Thus, social impact is their main success criterion [18,91]. Social impact can be broadly defined as "the process of transforming patterns of thought, behavior, social relationships, institutions, and social structure to generate beneficial outcomes for individuals, communities, organizations, society, and/or the environment beyond the benefits for the instigators of such transformations" ([2]; p. 1252). We argue that the positive effects of a successful provision of social impact are not limited to SEs' beneficiaries but also yield resources for job satisfaction and general MWB for social entrepreneurs themselves due to two reasons. First, regarding job satisfaction, success at one's job has been identified as a notable influence on job satisfaction (see Jalagat [92] for an overview and model). This is driven by the notion that people like performing actions that they are good at. Second, concerning general MWB, altruistically and benevolently helping people is the main driver of social entrepreneurs when choosing their career path [8,93]. Thus, founding a social enterprise that successfully provides social impact contributes to the fulfillment of one central life goal of social entrepreneurs. This way, it seems reasonable that job success effects for social entrepreneurs are not restricted to job satisfaction but spill over to positively affect their general MWB. Consequently, we derive the following hypothesis:

**H5.** *The successful provision of social impact has a positive effect on social entrepreneurs' job satisfaction and general MWB.*

(vi) Business model effects on social entrepreneurs' MWB and its resources

As outlined in Section 2.1., two main forms of SEs, for-profit and not-for-profit, exist. While the similarities (innovativeness, acting on competitive markets) and differences (for-profit SEs fuse the creation of social impact and profit, not-for-profit SEs acquire donations and public funding) are theoretically well-elaborated, to date, it remains unclear how these differences may impact SE-related outcomes such as job satisfaction and general MWB. On the one hand, differences regarding the levels of MWB and its resources could occur. To illustrate this point, for-profit social entrepreneurs' aspiration to financially self-sustain their enterprise is likely to pose an additional challenge compared to not-for-profit social entrepreneurs. This could negatively impact for-profit social entrepreneurs' MWB [89]. However, for the opposite, i.e., a positive MWB-effect, reasons can also be found, as not-for-profit social entrepreneurs are dependent on donations and funding; that is, they do not have the financing of their enterprise in their own hands. On the other hand, differences in the underlying mechanisms linking MWB and its resources could emerge. As an example, due to the hybridity of for-profit social enterprises, their enterprise success could be more facetted, as, next to social impact, financial elements are more pronounced compared to not-for-profit social entrepreneurs [20]. Thus, the effect of social impact on job satisfaction could be smaller.

Whereas, from a theoretical point of view, arguments exist implying that there could be differences in the levels of MWB and its resources, as well as underlying mechanisms, to the best of our knowledge, no empirical study examining this assumption exists to date. To shed light on this question, we explore these differences guided by the following research question (RQ):

**RQ1.** *Are there differences in (i) the levels of MWB and its resources and (ii) underlying mechanisms comparing for-profit and not-for-profit social entrepreneurs?*

As a summary, Figure 1 depicts our research model.

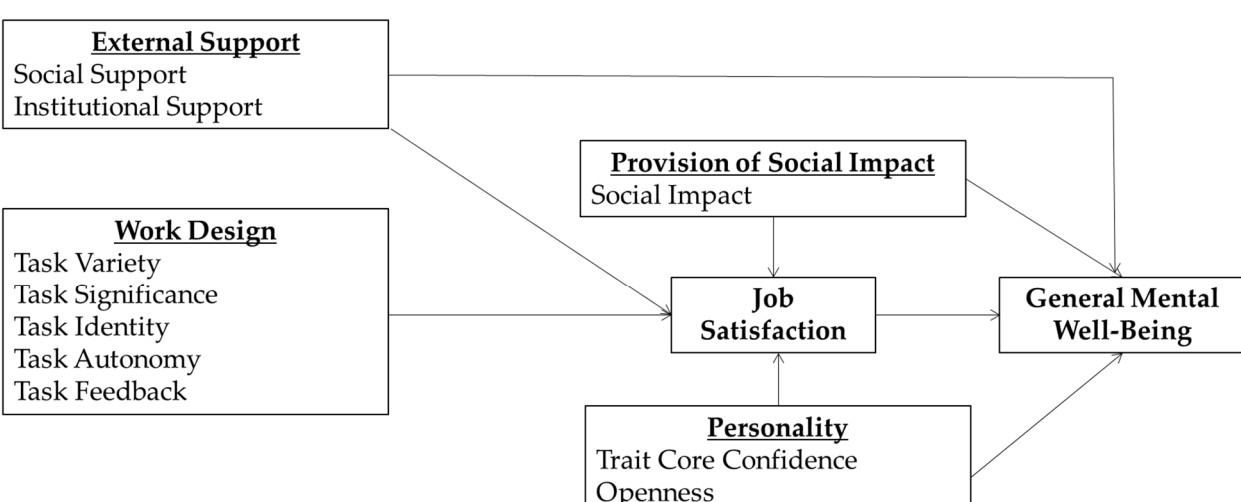

**Figure 1.** Research model.

### 3. Methods

#### *3.1. Sample Choice and Sampling Procedure*

To examine our hypotheses and research question, we collected primary data amongst social entrepreneurs in the two South African Provinces of Gauteng and Limpopo. This population was chosen for three main reasons. First, in general, South Africa has a rich and vivid landscape of social enterprises [94], and yet, compared to Western countries, it is massively underexplored in the behavioral and business sciences [95]. Second, the Provinces of Gauteng and Limpopo are considered very dynamic in their economic and social development and show particularly high levels of entrepreneurial activity [96,97]. Third, entrepreneurial activity in Gauteng and Limpopo is systematically recorded with a "Living List" generated by the Gordon Institute of Business Science (GIBS) in collaboration with the Bertha Centre for Social Innovation and Entrepreneurship. This list was used to identify potential survey participants.

The data-collection process took place between January 2021 and May 2021 as a part of a large-scale project surveying social entrepreneurial activity. As a representative sample of social entrepreneurs from the two provinces was targeted, a random sampling technique was employed. At first, an email system was used to reach potential respondents. However, due to a low response rate and incomplete responses, physical addresses supplied in the database were used to locate the respondents face-to-face. Data were collected using a personal, structured, and fully standardized interview with the social entrepreneurs based on a questionnaire crafted by the authors of this study. The interviews were conducted by a contracted research company.

#### *3.2. Sample Description*

Our sample was composed of 347 social entrepreneurs. A total of 72.6% led non-profit enterprises. The majority of enterprises have existed for 16–20 years (36.6%), followed by enterprises ranging between 11 and 15 years (26.2%), those ranging between 6 and 10 years (12.1%), those existing for more than 21 years (18.7%), and those existing for less than 6 years (6.3%). Most enterprises had 11–20 employees (37.5%), followed by less than 10 employees (32.6%) and more than 20 employees (30.0%). The majority of social entrepreneurs' ranged in age between 36 and 45 years (37.2%), followed by social entrepreneurs aged between 20 and 35 years (31.4%) and social entrepreneurs older than 46 years (31.4%). Of all social entrepreneurs included in our sample, 179 were female (51.6%) and 168 were male (48.4%).

*3.3. Measures*

(i) Personality

As personality-related resources for social entrepreneurs' MWB, TCC and the personal values openness and self-transcendence were measured. Trait core confidence, i.e., the dispositional and general belief of individuals in their ability to accomplish tasks, was assessed using the Trait Core Confidence Scale [74]. Participants rated six items, e.g., "I am confident in achieving my goals in life", on a 7-point Likert scale, ranging from 1 ("strongly disagree") to 7 ("strongly agree"). The scale's Cronbach's alpha was $\alpha = 0.78$. Openness values, suggesting high levels of free thinking, innovativeness, and the preparedness to take risks in one's life; and self-transcendence values, i.e., the personal disposition to benevolently help others and show altruistic behavior, were assessed using the Values Survey by Cable and Edwards [98]. On a five-point Likert scale (1 = not important at all; 5 = extremely important), participants rated six statements regarding the importance of actions, e.g., "Do something different every day" (openness) and "Making the world a better place" (self-transcendence). Cronbach's alphas were $\alpha = 0.75$ for openness and $\alpha = 0.80$ for self-transcendence.

(ii) Work Design

All elements of a good work design based on the Job Characteristics Model were operationalized using the Work Design Questionnaire by Morgeson and Humphrey [99] and a 7-point Likert scale, from 1 ("strongly disagree") to 7 ("strongly agree"). Task variety, i.e., the degree to which one's work encompasses different sets of actions and skills (e.g., "The job involves a great deal of task variety"); task identity, i.e., the degree to which one is involved in all working steps when performing a task (e.g., "The job allows me to complete work I start"); and task significance, i.e., the perception of performing work that makes sense and contributes to others' lives (e.g., "The job has a large impact on people outside my enterprise"), were measured with four items each. Task autonomy, the possibility of making one's own decisions at work, encompassed eight items, e.g., "The job allows me to plan how I do my work". Task feedback, i.e., the possibility to evaluate one's successful completion of a working task based on information imparted by the task, was covered by six items, e.g., "The job itself provides feedback on my performance". Cronbach's alphas of the five scales were in the range of $0.70 \leq \alpha \leq 0.84$.

(iii) External Support

External-support resources encompassed two elements. Social support, i.e., the extent to which important other people or peers appreciate and help when performing one's work, was adapted from Ajzen [100]. Participants rated four statements—e.g., "How much social support (e.g., positive feedback, help) for your enterprise do you receive by your friends?"—on a 7-point Likert scale (1 = no support at all; 7 = extremely high support). Cronbach's alpha was $\alpha = 0.75$. Institutional support, i.e., the extent to which social entrepreneurs receive money and educational or other resources by national institutions, e.g., the government, was assessed with three items from a questionnaire developed by Bloom and Smith [91], on a 7-point Likert-scale (1 = almost never to 7 = very often). An example item read, "I have been successful at getting government agencies and officials to provide financial support for my efforts as an entrepreneur" ($\alpha = 0.94$).

(iv) Provision of Social Impact

Social impact, which is broadly defined as the creation of social value for beneficiaries in need, was assessed with a scale by Bloom and Smith [91]. On a 7-point Likert scale, from 1 ("not successful") to 7 ("very successful"), participants indicated how successful they were in the last three years accomplishing their social mission (e.g., "I have made significant progress in alleviating the problem my enterprise addresses"). Cronbach's alpha was $\alpha = 0.82$.

(v) Mental Well-Being

For the assessment of job-specific MWB, i.e., job satisfaction and general MWB, two different scales were used. Job satisfaction, expressing the extent to which people like their job and enjoy performing their work, was measured with the Job Satisfaction Ques-

tionnaire [101]. The scale ($\alpha$ = 0.89) consists of five items on a 7-point Likert scale, from 1 ("strongly disagree") to 7 ("strongly agree"), and an example item was "Most days I am enthusiastic about my work". General MWB, i.e., the ability to actively participate in society and live a happy life, was operationalized with four items taken from the Questionnaire for Eudaimonic Well-Being [102]. Participants rated statements such as "I feel best when I'm doing something worth investing a great deal of effort in" on a 7-point Likert-scale, from strongly disagree (= 1) to strongly agree (= 7). Cronbach's alpha was $\alpha$ = 0.69.

(vi) Control variables

As control variables, we included social entrepreneurs' age and gender. When reviewing the pertinent MWB literature, we noted that these two sociodemographic variables are frequently included to account for their proven effects on MWB [103,104].

### 3.4. Statistical Analyses

When analyzing our sample, we followed a four-step procedure. First, to gain an overview of the data, we computed the means, standard deviations, and intercorrelations of all study variables. Second, as pre-analyses, we examined outliers by using Mahalanobis distances [105] and univariate and multivariate data normality [106]. Third, when testing our hypotheses, a structural equation model (SEM) was crafted in line with our research model depicted in Figure 1. The model was composed of manifest variables and featured all intercorrelations. Fourth, to explore our research question, we used (i) a Mann–Whitney U test for mean differences in MWB and its resources [107] and (ii) multigroup structural equation modeling to explore differences in MWB-related mechanisms. In both analyses, we compared for-profit and not-for-profit entrepreneurs.

## 4. Results

### 4.1. Data Overview

Table 1 shows the means, standard deviations, and intercorrelations of all variables. Regarding our two MWB variables, several significant correlations can be observed. For job satisfaction, all intercorrelations with personality, work design, external support, and social impact resources emerged as significant. They range between $r$ = 0.26 ($p < 0.01$) for task variety and $r$ = 0.86 ($p < 0.01$) for task significance. Furthermore, correlations with the control variables gender ($r$ = −0.15; $p < 0.01$) and age ($r$ = 0.36; $p < 0.01$) can be remarked, suggesting that males (coded as "1"; compared to females coded as "2") and older people (coded as age in years) report higher job-satisfaction scores. For general MWB, except for institutional support, all intercorrelations are significant. They range between $r$ = 0.21 ($p < 0.01$) for social support and $r$ = 0.68 ($p < 0.01$) for TCC. The significant correlation with age ($r$ = 0.18; $p < 0.01$) suggests that older people report higher general MWB levels. Moreover, a significant correlation with job satisfaction emerged ($r$ = 0.35; $p < 0.01$).

**Table 1.** Means, standard deviations (SDs), and intercorrelations of all study variables (N = 347).

| Variables | Mean (SD) | 1 | 2 | 3 | 4 | 5 | 6 | 7 | 8 | 9 | 10 | 11 | 12 | 13 | 14 | 15 |
|---|---|---|---|---|---|---|---|---|---|---|---|---|---|---|---|---|
| 1. Gender | 1.52 (0.50) | - | | | | | | | | | | | | | | |
| 2. Age | 2.16 (1.11) | −0.20 ** | - | | | | | | | | | | | | | |
| 3. General MWB | 6.42 (0.67) | 0.02 | 0.18 ** | - | | | | | | | | | | | | |
| 4. Job Satisfaction | 5.01 (1.67) | −0.15 ** | 0.36 ** | 0.35 ** | - | | | | | | | | | | | |
| 5. Task Variety | 6.44 (0.65) | −0.05 | 0.20 ** | 0.54 ** | 0.26 ** | - | | | | | | | | | | |
| 6. Task Identity | 6.08 (0.86) | −0.14 ** | 0.31 ** | 0.57 ** | 0.68 ** | 0.43 ** | - | | | | | | | | | |
| 7. Task Significance | 5.62 (1.25) | −0.22 ** | 0.39 ** | 0.25 ** | 0.86 ** | 0.19 ** | 0.64 ** | - | | | | | | | | |
| 8. Task Feedback | 5.32 (1.56) | −0.16 ** | 0.37 ** | 0.23 ** | 0.85 ** | 0.19 ** | 0.65 ** | 0.89 ** | - | | | | | | | |
| 9. Task Autonomy | 5.16 (1.14) | −0.16 ** | 0.25 ** | 0.24 ** | 0.77 ** | 0.23 ** | 0.62 ** | 0.80 ** | 0.80 ** | - | | | | | | |
| 10. Self-Transcendence | 4.68 (0.47) | −0.08 | 0.22 ** | 0.66 ** | 0.46 ** | 0.50 ** | 0.61 ** | 0.37 ** | 0.35 ** | 0.38 ** | - | | | | | |
| 11. Trait Core Confidence | 6.41 (0.71) | −0.04 | 0.14 ** | 0.68 ** | 0.34 ** | 0.56 ** | 0.56 ** | 0.27 ** | 0.23 ** | 0.33 ** | 0.74 ** | - | | | | |
| 12. Openness | 4.71 (0.43) | −0.05 | 0.20 ** | 0.59 ** | 0.39 ** | 0.49 ** | 0.54 ** | 0.31 ** | 0.29 ** | 0.35 ** | 0.75 ** | 0.66 ** | - | | | |
| 13. Institutional Support | 2.93 (1.84) | −0.03 | 0.02 | 0.00 | 0.28 * | −0.02 | 0.17 ** | 0.32 ** | 0.33 * | 0.42 ** | −0.01 | −0.02 | −0.01 | - | | |
| 14. Social Support | 5.11 (1.22) | −0.13 * | 0.30 ** | 0.21 * | 0.76 ** | 0.22 ** | 0.55 ** | 0.75 ** | 0.80 ** | 0.66 ** | 0.34 ** | 0.21 ** | 0.26 ** | 0.42 ** | - | |
| 15. Social Impact | 5.96 (0.92) | −0.16 ** | 0.34 ** | 0.56 ** | 0.76 ** | 0.48 ** | 0.79 ** | 0.73 ** | 0.73 ** | 0.68 ** | 0.69 ** | 0.61 ** | 0.63 ** | 0.20 ** | 0.64 ** | - |

Note: MWB = mental well-being. Gender: 1 = male; 2 = female. * $p < 0.05$; ** $p < 0.01$.

### 4.2. Pre-Analyses

To identify outliers in our data, we calculated the Mahalanobis distances ($D^2$) for each social entrepreneur in our original sample. $D^2$-scores indicate the distance between observations—in our case, the responses of social entrepreneurs—and, thereby, it can be checked whether single observations should be treated as outliers, as the assumption of equidistance to other observations is not met. Following the recommendations by Penny [108], we had to exclude three social entrepreneurs from our analyses, as their Mahalanobis distances violated the proposed critical ratio for our sample. Thus, the total sample was reduced to $N = 344$.

In our examination of the univariate normality, i.e., the normal distribution of every variable included in our research model, the skewness and kurtosis values were computed [106]. Comparing the results to the commonly accepted thresholds of |2| for kurtosis and |7| for skewness (scores exceeding these cutoff-values would be treated as non-normal [109,110]), we found that the majority of variables fulfilled the criteria of univariate normality. For skewness, task variety (|2.94|), TCC (|2.59|), self-transcendence (|2.08|), and general MWB (|2.34|) displayed scores slightly over the proposed threshold. For kurtosis, task variety (|10.92|), openness (|8.96|), and self-transcendence (|8.63|) exceeded the threshold. When investigating multivariate normality, we calculated Mardia's critical ratio. According to Yung and Bentler [111], scores should not exceed the threshold of five. In our analysis, the critical ratio was notably higher, indicating that there is no multivariate normality in the data (68.29). However, as Byrne [106] remarks, failure to meet the strict regulations for multivariate normality is common in non-simulated data. Furthermore, a Monte Carlo simulation study by Hallow [112] could show that in structural equation modeling, the estimate and standard error bias caused by multivariate non-normality is neglectable. Nevertheless, accounting for the violations of some variables regarding univariate normality and to take multivariate non-normality into account, we followed the recommendations by Yung and Bentler [111] and West, Finch, and Curran [110] and applied a Maximum Likelihood Bootstrapping procedure as an effective method to handle non-normality [113].

### 4.3. Hypotheses Examination

All direct effects for our SEM are displayed in Table 2.

**Table 2.** Summary of SEM results (direct effects) for the complete sample of South African social entrepreneurs ($N = 344$) based on our research model.

| Resource | Variable | Variable Effect on | |
|---|---|---|---|
| | | **Job Satisfaction** | **General MWB** |
| Job-Specific MWB | Job Satisfaction | - | 0.02 |
| | Trait Core Confidence | −0.02 | 0.38 ** |
| Personality | Openness | 0.01 | 0.10 |
| | Self-Transcendence | 0.05 | 0.21 * |
| | Task Variety | −0.01 | - |
| | Task Significance | 0.36 ** | - |
| Work Design | Task Identity | 0.05 | - |
| | Task Autonomy | 0.10 + | - |
| | Task Feedback | 0.20 ** | - |
| External Support | Social Support | 0.15 ** | −0.10 |
| | Institutional Support | 0.05 | 0.02 |
| Provision of Social Impact | Social Impact | 0.13 * | 0.16 |

Note: SEM = structural equation modeling; MWB = mental well-being. Path Coefficients are displayed as standardized values (β) and based on 2000 bootstraps, all intercorrelations and age and gender as control variables were included in computation; + = $p < 0.10$; * = $p < 0.05$; ** = $p < 0.01$.

Regarding H1, predicting positive effects of job satisfaction on general MWB, no significant effect was found; thus, H1 was rejected. H2.1–2.3, which focused on personality resources for job satisfaction and general MWB, received mixed support. While H2.1 and H2.3, which postulated the positive effects of TCC and self-transcendence, were supported for general MWB ($\beta_{TCC}$ = 0.38, $p < 0.01$; $\beta_{self\text{-}transcendence}$ = 0.21, $p < 0.05$), no effects on job satisfaction were found. Hypothesis H2.2, which focused on openness, was fully rejected, as neither a significant effect for job satisfaction nor general MWB emerged. Concerning work design, calculations yielded positive direct effects of task significance ($\beta$ = 0.36, $p < 0.01$) and task feedback ($\beta$ = 0.20, $p < 0.01$) on job satisfaction, thus supporting H3. Moreover, a marginally significant effect of task autonomy was found ($\beta$ = 0.10, $p < 0.10$). However, the other two work-design features were non-significant. Furthermore, no indirect effects on MWB emerged. Thus, H3 only received partial support. The examination of H4.1 and H4.2 on external support resources uncovered a positive effect of social support on job satisfaction ($\beta$ = 0.15, $p < 0.01$), yet no effect on general MWB was detected. No significant effect was found for institutional support. Consequently, while H4.1 was partially supported, H4.2 had to be rejected. For social impact, a positive effect on job satisfaction was found ($\beta$ = 0.13, $p < 0.01$), which is in line with H5. However, no effect on general MWB emerged, thus resulting in partial support for our hypothesis.

Regarding overall model fit, the three most common fit indices, i.e., chi-square value ($\chi 2$ [6] = 28.64, $p < 0.01$), comparative fit index (*CFI* = 0.99), and root mean square error of approximation (*RMSEA* [90% CI] = [0.07; 0.14]), are all in a good to very good range [114]. This indicates that our theoretical model appropriately fits the empirical data. This is also supported considering the high levels of variance explained for job satisfaction ($R^2$ = 0.82, $p < 0.01$) and general MWB ($R^2$ = 0.54, $p < 0.01$).

### 4.4. Research Question Examination

To examine our research question on differences comparing for-profit and not-for-profit social entrepreneurs, we applied two different statistical procedures:

First, when investigating the differences in the *levels* of MWB and its resources, we conducted a Mann–Whitney U test. On the one hand, this non-parametric test was very robust against violations of data normality that occurred in some of our variables (see Section 4.2). On the other hand, our data do not fulfill the assumption of variance homogeneity as a requirement for several mean difference tests such as analyses of variance [115]. The Mann–Whitney U test can deal with variance heterogeneity; thus, this test was chosen [107]. As can be seen in Table 3, several significant differences emerged. For-profit social entrepreneurs report higher levels of work-design resources (with the exception of task variety) and more external support. Furthermore, they provide a greater social impact and score higher on job satisfaction. Not-for-profit social entrepreneurs indicate higher levels of openness, self-transcendence, and TCC (marginally significant). Notably, no differences regarding general MWB occurred. The effect sizes of the significant coefficients were in the range $0.10 \leq r \leq 0.37$.

Second, to explore differences in MWB *mechanisms*, multigroup structural equation modeling (MSEM) was applied. Thus, we used our research model and ran it separately for two groups (for-profit and not-for-profit social entrepreneurs). As coefficients for both groups are calculated simultaneously, this method is particularly suitable to avoid alpha-error inflations and thus is frequently applied (see Chipeta, Kruse, and Venter [76] for an example in the SE context). We also used 2000 bootstrapping samples in this calculation. Tables 4 and 5 summarize the results. Checking the two tables, we see that similarities and differences emerged. Amongst *for-profit social entrepreneurs*, both external support variables were found to be significant for job satisfaction. Moreover, a marginally significant effect of the work-design variable task variety occurred. Regarding general MWB, the personality variable TCC and social impact emerged as marginally significant. *Not-for profit social entrepreneurs'* job satisfaction was influenced by the personality variable self-transcendence, the work-design variables task significance and task feedback, and social impact. Their

general MWB was influenced by the personality variables TCC and self-transcendence and the external support variable social support. In both groups, no significant indirect effects occurred. The total model fit of our MSEM was, following Hooper, Coughlan, and Mullen [114] very good to excellent ($\chi^2$ [12] = 34.74, $p < 0.01$; *CFI* = 0.99; *RMSEA* [90% CI] = [0.05; 0.10]).

**Table 3.** Summary of Mann–Whitney U test results for mean differences comparing for-profit (*n* = 92) and not-for-profit (*n* = 252) social entrepreneurs.

|  | Variable | $|\Delta_M|$ | *z* | *r* | Higher Amongst |
|---|---|---|---|---|---|
| MWB | Job Satisfaction | 1.32 | −5.93 ** | 0.32 | FP |
|  | General MWB | −0.18 | −1.14 | 0.06 | n.s. |
| Personality | Trait Core Confidence | 0.18 | −1.84 + | 0.10 | NFP |
|  | Openness | 0.05 | −2.87 ** | 0.15 | NFP |
|  | Self-Transcendence | −0.09 | −2.47 * | 0.13 | NFP |
| Work Design | Task Variety | 0.10 | −0.58 | 0.03 | n.s. |
|  | Task Significance | 1.07 | −6.90 ** | 0.37 | FP |
|  | Task Identity | 0.26 | −4.97 ** | 0.27 | FP |
|  | Task Autonomy | 0.95 | −6.92 ** | 0.37 | FP |
|  | Task Feedback | 0.26 | −6.22 ** | 0.34 | FP |
| External Support | Social Support | 0.83 | −5.58 ** | 0.30 | FP |
|  | Institutional Support | 1.10 | −3.42 ** | 0.18 | FP |
| Provision of Social Impact | Social Impact | 0.32 | −4.69 ** | 0.25 | FP |

Note: MWB = mental well-being; $\Delta_M$ = mean difference; *z* = Mann–Whitney U test value; *r* = Mann–Whitney U test effect size calculated as $r = \left|\frac{z}{\sqrt{N}}\right|$; FP = for-profit social entrepreneur; NFP = not-for-profit social entrepreneurs; n.s. = not significant; + = $p < 0.10$; * = $p < 0.05$; ** = $p < 0.01$.

**Table 4.** Summary of MSEM-results (direct effects) of the for-profit (*n* = 92) South African social entrepreneurs based on our research model.

| Resource | Variable | Variable Effect on | |
|---|---|---|---|
|  |  | **Job Satisfaction** | **General MWB** |
| Job-Specific MWB | Job Satisfaction | - | 0.14 |
| Personality | Trait Core Confidence | 0.24 | 0.28 + |
|  | Openness | 0.28 | 0.11 |
|  | Self-Transcendence | −0.11 | −0.03 |
| Work Design | Task Variety | −0.19 + | - |
|  | Task Significance | 0.06 | - |
|  | Task Identity | 0.06 | - |
|  | Task Autonomy | 0.02 | - |
|  | Task Feedback | 0.12 | - |
| External Support | Social Support | 0.19 * | 0.08 |
|  | Institutional Support | −0.13 * | 0.04 |
| Provision of Social Impact | Social Impact | 0.25 | 0.35 + |

Note: MSEM = multigroup structural equation modeling; MWB = mental well-being. Path Coefficients are displayed as standardized values (β) and based on 2000 bootstraps. All intercorrelations and age and gender as control variables were included in computation; $R^2_{\text{Job satisfaction}}$ = 0.71 **; $R^2_{\text{General MWB}}$ = 0.75 **; + = $p < 0.10$; * = $p < 0.05$; ** = $p < 0.01$.

**Table 5.** Summary of MSEM-results (direct effects) for not-for-profit (*n* = 252) South African social entrepreneurs based on our research model.

| Resource | Variable | Variable Effect on | |
| --- | --- | --- | --- |
| | | Job Satisfaction | General MWB |
| Job-Specific MWB | Job Satisfaction | - | 0.14 |
| | Trait Core Confidence | −0.02 | 0.30 ** |
| Personality | Openness | 0.02 | 0.04 |
| | Self-Transcendence | 0.08 * | 0.29 ** |
| | Task Variety | 0.05 | - |
| | Task Significance | 0.38 ** | - |
| Work Design | Task Identity | 0.07 | - |
| | Task Autonomy | 0.08 | - |
| | Task Feedback | 0.20 * | - |
| External Support | Social Support | 0.07 | −0.20 + |
| | Institutional Support | −0.02 | 0.03 |
| Provision of Social Impact | Social Impact | 0.13 ** | 0.04 |

Note: MSEM = multigroup structural equation modeling; MWB = mental well-being. Path Coefficients are displayed as standardized values (β) and based on 2000 bootstraps; all intercorrelations and age and gender as control variables were included in computation; $R^2_{\text{Job satisfaction}}$ = 0.84 **; $R^2_{\text{General MWB}}$ = 0.35 **; + = $p < 0.10$; * = $p < 0.05$; ** = $p < 0.01$.

## 5. Discussion

Taking a psychological view on resources, the current study explored what keeps social entrepreneurs happy despite several job-related challenges and stressors. Doing so, we integrated personality, work design, external support, and social impact as resources in our research model to investigate their effects on job satisfaction and general mental well-being (MWB). Using a representative sample of South African social entrepreneurs from Gauteng and Limpopo Provinces, structural equation modeling (SEM) was applied to check our hypotheses. Furthermore, we shed light on the research question whether for-profit and not-for-profit social entrepreneurs differ in (i) their levels of MWB and its resources and (ii) underlying mechanisms by conducting Mann–Whitney U tests and multigroup SEM.

The first hypothesis, H1, postulating a positive effect of job satisfaction, i.e., job-specific MWB on social entrepreneurs' general MWB, was not confirmed. This seems surprising, considering that social entrepreneurs' work is a central element in their lives, as they usually work very long hours and mostly chose this career option to reach their deeply rooted altruistic motivations. However, in light of the significant correlation of both constructs (*r* = 0.35, *p* < 0.01; see Table 1) and the several significant correlations of job satisfaction with our proposed MWB resources, the reason for our finding could lie in a statistical effect referred to as "multicollinearity" [116]. If this effect occurs, significant bivariate correlations do not translate to SEM effects, as other variables (in our case, the MWB resources) explain so much variance that the share explained by job satisfaction alone is too low to result in a significant coefficient. Thus, the rejection of H1 could be rooted in statistical multicollinearity.

H2.1–2.3 focused on the effects of personality resources on job satisfaction and general MWB. While, in line with our hypotheses, significant and positive effects of TCC and self-transcendence on general MWB were found, no effect for openness emerged. Regarding job satisfaction, no personality effects were found. The finding that personality resources did not affect job satisfaction could be explained by the model of distance and proximity by Kanfer [82]. The model states that personality traits are, on a conceptual level, not as close to job satisfaction as job-specific variables, e.g., work-design characteristics. Thus, their effects should be lower. A study by Judge et al. [117] empirically supported this assumption, and, in the light of several work design effects on job satisfaction (see paragraph on hypothesis H3), this logic also seems feasible for social entrepreneurs.

When examining H3, which focused on work-design effects on job satisfaction and general MWB, task significance and task feedback emerged as significant predictors for

social entrepreneurs' job satisfaction. Moreover, a marginally significant effect for task autonomy occurred. This provides additional evidence for the importance of good work design postulated in the Job Characteristics Model [79] and highlights the beneficial effects of integrating work design and (social) entrepreneurship research for the explanation of job satisfaction [81]. However, in contrast to our hypothesis, no indirect effects on general MWB emerged. One reason for this could be that other variables, such as work performance—i.e., in our case, the provision of social impact—could serve as better mediators of wok design effects.

H4.1 and H4.2, encompassing the effects of social and institutional support on job satisfaction and general MWB, yielded a positive effect of social support on job satisfaction, as is in line with findings in other occupations [80]. No effects were found for general MWB. The finding that institutional support was insignificant regarding job satisfaction and general MWB shows that, as previous contradictory research suggests, the effect of institutional support is not general and dependent on different contextual factors such as the sector social entrepreneurs operate in (see Kruse [3] for a summary and empirical evidence).

H5, suggesting the positive effect of social impact provision, was supported for job satisfaction but not for general MWB. Possibly, social impact, i.e., the main criterion for social enterprise success, is predominantly seen as a job-specific resource, and spillover effects to general MWB do not occur. However, considering the significant bivariate correlation, similar to the job-satisfaction–general-MWB relationship ($r = 0.56$, $p < 0.01$; see Table 1), multicollinearity could also be responsible for this finding.

Concerning our research question, we find remarkable differences when comparing for-profit and not-for-profit social entrepreneurs. Our data suggest that for-profit social entrepreneurs have a better work design, have more external support, and provide more social impact. Furthermore, they report higher levels of job satisfaction. In contrast, not-for-profit social entrepreneurs consider themselves to be in possession of more personality resources. As for-profit social entrepreneurs face the enormous challenge of balancing social and financial value creation, it seems reasonable that they are in need of more resources to keep their enterprise running. This could explain why they indicate higher resource levels in work design and external support. Moreover, providing more social impact could result in higher job satisfaction amongst for-profit social entrepreneurs. This supports claims that, despite the inherent risks of running a hybrid social enterprise, taking this risk can pay off for the SE's beneficiaries and the social entrepreneur him/herself [4]. The higher personality resources amongst not-for-profit social entrepreneurs could originate from a more distinct focus on social-value creation in this form of SE, resulting in lower financial aspirations and, consequently, less ambiguity. Thus, the benefit directly drawn from personality dispositions could be bigger as, e.g., pragmatic cognitively based decisions to secure financial independence on the expense of social mission accomplishment occur less frequently [43]. Regarding mechanisms, both for-profit and not-for-profit social entrepreneurs' MWB benefits from high levels of TCC and the provision of social impact. Remarkably, social impact has a (marginally) significant effect on general MWB for for-profit and a significant effect on job satisfaction for not-for profit social entrepreneurs. This could imply that social value creation amongst not-for-profit social entrepreneurs has a higher importance in a job context. In fact, despite the underlying altruistic motives of all social entrepreneurs, one essential business model difference is the full social value focus of not-for-profit compared to for-profit social entrepreneurs [8,11]. As a result, social-impact creation could have a higher weight when rating job satisfaction compared to for-profit social entrepreneurs, who probably consider social *and* financial value creation. The significant effect of the personality resource self-transcendence on not-for-profit social entrepreneurs' general MWB provides additional support for their deeply rooted social motives. It is worth noting that, amongst for-profit social entrepreneurs, the effects of external support seem differential. While social support positively affects job satisfaction, there is a negative coefficient for institutional support. This negative effect could originate

from a potential conflict between the institutional help provided and the goal to keep the social enterprise financially independent and, thus, free from political influences [118].

### 5.1. Implications for Research and Practice

Our study has several implications for researchers and practitioners in the field:

First, our study is amongst the very first empirically exploring social entrepreneurs' job satisfaction and general MWB *resources*. This way, we widen the scope and overcome the largely restricted perspective of emphasizing stressors and risk factors that social entrepreneurs suffer from.

Second, as our empirical research model, which is composed of personality, work design, external support, and social impact resources, has an overall good empirical fit, future research can build on our model in two ways. On the one hand, adding other variables in the clusters (e.g., the Big Five Personality Traits in the personality resources cluster) could lead to a more fine-grained picture of which variables in the resource clusters have the highest predictive validity. On the other hand, as we consider our framework to be an open framework, adding more resource clusters (e.g., a human capital or entrepreneurial experience cluster [83]) could add to the explanatory power of the model.

Third, the link between job-specific and general MWB of social entrepreneurs requires more research exploring how strong the relationship between both constructs is. Furthermore, more effort has to be invested to identify and empirically investigate potential mediators and moderators. As we did not find significant mediation effects in our model, including new paths, linking work design and social impact seems promising.

Fourth, significant differences comparing not-for-profit and for-profit social entrepreneurs emerged. As this indicates that theoretically proposed business model differences on the organizational level also become manifest in quantifiable person-related outcomes, such as personality or work design, we encourage scholars to acknowledge and further explore these differences.

Fifth, as SE is widely considered an important tool to sustainably tackle challenges, such as poverty or social inequality, knowledge about resources that help social entrepreneurs to accomplish their tasks is essential. Our work identifies several resources that are subject to change like work design characteristics and unearths the positive effects of social support for social entrepreneurs' job satisfaction. These findings could be used to design governmental support programs whose scope combines "traditional" insights (e.g., on how to administratively manage one's business or funding information) with expert workshops offering help on good work design or peer networks for social entrepreneurs.

### 5.2. Limitations

Notwithstanding the contributions, the following limitations of our work should be considered:

First, personality, work design, external support, and the provision of social impact were treated as resources of job satisfaction and general MWB. However, as our data are cross-sectional, we cannot account for reverse causality, i.e., the direction of the relationship between our resources and the MWB outcomes cannot be determined. Future work should acquire longitudinal data to make up for this shortcoming.

Second, our sample is limited to the context of two South African Provinces. Thus, findings can neither be generalized regarding the whole country nor other geographical areas, e.g., in Europe or North America, in which social entrepreneurs operate under considerably different circumstances [94].

Third, due to data protection and anonymity concerns, we were restricted regarding the acquisition of the sociodemographic data of social entrepreneurs. This is why only age and gender could be assessed. However, other sociodemographics, such as the educational/academic background, could also be important variables predicting social entrepreneurial MWB.

Fourth, our study is based on self-reported data, as is common in the field of MWB. However, future research should try to complement subjective perceptions with (relatively) objective indicators of financial success (earnings, donations, etc.).

Fifth, despite having a sufficient number of not-for-profit and for-profit social entrepreneurs to statistically compare both groups, the share of not-for-profit entrepreneurs was notably higher. Subsequent studies should try to acquire balanced samples to make comparisons of different business models more robust.

Sixth, using multigroup SEM as a method to compare not-for-profit and for-profit social entrepreneurs has numerous advantages, such as the avoidance of alpha-error inflation, as all calculations are conducted in one single model (cf. Byrne [106]). Future research, however, should also consider applying Compositional Data Analysis based on log ratios [119]. This method is less susceptible to spurious correlations and can help to assess the robustness of the current findings in subsequent studies.

## 6. Conclusions

Examining a large sample of South African social entrepreneurs, the current study took a psychological view and explored how four resource clusters (personality, work design, external support, and the provision of social impact) affect job satisfaction and general mental well-being, using structural equation modeling. Our results yielded positive effects of the personality dispositions' trait core confidence and self-transcendence on social entrepreneurs' general mental well-being and showed the importance of good work design (particularly task significance and task feedback), social support, and social impact creation for job satisfaction. Furthermore, we unearthed differences comparing for-profit and not-for-profit social entrepreneurs regarding their resources and job satisfaction levels. Despite notable limitations, such as the usage of cross-sectional data not allowing causal conclusions and a limited generalizability of our sample, our work has several important implications. Given the good empirical fit, future research can build on our research model and add (i) more variables in the proposed clusters and (ii) more clusters to improve its explanatory power. Practitioners can benefit from our findings in promoting (nascent) social entrepreneurs' job satisfaction and mental well-being, e.g., by offering advice on good work design or implementing peer networks offering social support.

**Author Contributions:** Conceptualization, P.K. and E.M.C.; methodology, P.K., E.M.C. and I.U.; data analysis, P.K. and I.U.; writing and manuscript preparation, P.K., E.M.C. and I.U. All authors have read and agreed to the published version of the manuscript.

**Funding:** The research project was funded by National Research Fund in collaboration with the University of the Witwatersrand.

**Informed Consent Statement:** Informed consent was obtained from all subjects involved in the study.

**Data Availability Statement:** The study data are available upon reasonable request from the correspondence author.

**Conflicts of Interest:** The authors declare no conflict of interest.

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
