# Peer review of "What Keeps Social Entrepreneurs Happy? Exploring Personality, Work Design, External Support, and Social Impact as Resources of Social Entrepreneurs’ Mental Well-Being"

_sustainability, doi:10.3390/su15054109_

Round 1

Reviewer 1 Report

The article is very interesting, well structured, with an interesting and relevant topic, and with an appropriate and well-implemented methodology. For these reasons, the article is an excellent candidate for publication. 

However, I think there is a major concern to be addressed by the authors. As far as I know, the RBV is an organizational theory so I do not understand the usefulness of its application in this study. One could still consider that they were studying organizational variables, but looking at the measures (for example: Personality measured with items such as "'I am confident in achieving my goals in life"), I have no doubt that these are personal/psychological constructs, so they cannot be considered organizational resources. Eventually, other constructs may fit into the concept of resources, although when the questions are asked in the first person many doubts remain: 

is an entrepreneur an organization?

Going forward, at the same time, I notice that the entrepreneur is being considered as an employee of his or her organization (for example when work design is asked). So, from which perspective are we analyzing? The entrepreneur's, the organization's or the employee's?

And how can all this be considered as 'resources'? Is Personality a resource?

Thus, I strongly recommend that the authors bring further clarification of the correct application of this theory, or alternatively, resort to another more appropriate theory.

In answering this point and possibly not providing adequate justification for the use of RBV, I recommend revising the paper in terms of discussion, conclusions, and the references to RBV in the introduction and abstract.

Author Response

Dear Reviewer 1,

Thank you so much for all the time, work, and dedication invested in our manuscript.

Please see attachment for our responses to your comments.

Take care and all the best,

The Authors 

Reviewer 2 Report

Dear authors, I found the study very interesting and very robust. I think that the authors could have even sectioned the analyses to make more publications. I also like the way the authors have justified the hypotheses and research questions, and how they take them up in the discussion and in the practical implications. Congratulations.

I would like to make a number of minor observations/comments that I think may help the authors to improve the quality of such good work.

1 I have noticed that the references you use to support most of your work are outdated, I do not know if it is part of the same thing that you mention that little has been done to study social entrepreneurship and the mental wellbeing of entrepreneurs. Only 8 of the 85 references used in the manuscript are from 2020 onwards.

2 Lines 126 to 137 . In these lines authors mention the interest that social entrepreneurship has gained in recent years and how it has been addressed by different disciplines (i.e., economics, sociology and psychology). In this sense, this argument has been exemplified with 3 disciplines and only two references.

3 Lines 126 to 137 . I think it would be important to mention how it has been approached by studies in education. Especially those on the development of social entrepreneurship competencies in university students to face today's complex challenges. Especially to combat inequality and poverty, as mentioned in the text. Moreover I think it can help the authors to reinforce what they have placed in line 704 - 706. In this sense, the authors can use as a reference a publication of this same journal or another placed in Social Sciences of the same publisher and updated (2022). Authors can also refer to other studies in case they feel that these are not useful to them.

Vázquez-Parra, J.C.; Cruz-Sandoval, M.; Carlos-Arroyo, M. Social Entrepreneurship and Complex Thinking: A Bibliometric Study. Sustainability 2022, 14, 13187. https://doi.org/10.3390/su142013187

Cruz-Sandoval, M.; Vázquez-Parra, J.C.; Alonso-Galicia, P.E. Student Perception of Competencies and Skills for Social Entrepreneurship in Complex Environments: An Approach with Mexican University Students. Soc. Sci. 2022, 11, 314. https://doi.org/10.3390/socsci11070314

4 Section 3.2 Description of the sample. I leave this for your consideration, but I think it would be good to specify both in absolute values and in percentage the gender of your sample of entrepreneurs. I.e. you mention that 51.6% are women. But sometimes the reader is lazy, it would be good to complement with absolute values and to include the male part of the sample both in absolute values and in percentages.

5 Section 3.2 Description of the sample. I do not know if you have the information. It would be nice whether it is possible to include  the academic background of the entrepreneurs by discipline.

6 Section 3.4. Statistical Analysis and Section 5 Results. I have found that the expected results in some of the associations between variables resulting from the multi-group structural equation modelling have surprised the authors by not finding a correlation/association. In this regard, I invite the authors to consider Compositional Data Analysis, based on log-ratios, for future research. This methodology focuses on the relative information of the data avoiding spurious correlations (and other problems) in data characterised by the constant sum constraint (nature of the dataset).

7 Perhaps section 6 on line 731 should be called conclusions instead of Summary.

8 Finally, I liked the way you have presented the discussion section, taking up each of your hypotheses and answering the research question. In this sense, they have responded to doubts that arose while I was reading their results. I also liked the way you presented the section on the implications and limitations of your research.

Author Response

Dear Reviewer 2,

Thank you so much for all the time, work, and dedication invested in our manuscript.

Please see attachment for our responses to your comments.

Take care and all the best,

The Authors 

Round 2

Reviewer 1 Report

Thank you for responding to all the topics.